# Adaptive Rank Allocation: Speeding Up Modern Transformers with RaNA Adapters

**Roberto Garcia**[†][*]**, Jerry Liu**[†]**, Daniel Sorvisto**[†] **and Sabri Eyuboglu**[‡]
[†]Institute of Computational and Mathematical Engineering, Stanford University
[‡]Department of Computer Science, Stanford University

## Abstract

Large Language Models (LLMs) are computationally intensive, particularly during inference. Neuron-adaptive techniques, which selectively activate neurons in Multi-Layer Perceptron (MLP) layers, offer some speedups but suffer from limitations in modern Transformers. These include reliance on sparse activations, incompatibility with attention layers, and the use of costly neuron masking techniques. To address these issues, we propose the Adaptive Rank Allocation framework and introduce the Rank and Neuron Allocator (RaNA) adapter. RaNA adapters leverage rank adapters, which operate on linear layers by applying both low-rank matrix decompositions and adaptive masking to efficiently allocate compute without depending on activation sparsity. This enables RaNA to be generally applied to MLPs and linear components of attention modules, while eliminating the need for expensive maskers found in neuron-adaptive methods. Notably, when compared to neuron adapters, RaNA improves perplexity by up to 7 points and increases accuracy by up to 8 percentage-points when reducing FLOPs by $\sim 44\%$ in state-of-the-art Transformer architectures. These results position RaNA as a robust solution for improving inference efficiency in modern Transformer architectures.

## 1 Introduction

As Large Language Models (LLMs) have grown in popularity and size, they have begun consuming a non-trivial amount of compute and time for training and inference (Kim et al. (2023), Pope et al. (2022)). Adaptive compute methods seek to speed up the inference stage of Transformers (Vaswani et al. (2023)), the *de facto* LLM architecture, by identifying and avoiding redundant computations to save I/O and floating-point operations (FLOPs). Commonly, these methods apply neuron adapters to the Multi Layer Perceptron (MLP) layers of the Transformer architecture, which dynamically ignore neurons depending on the input of the layer (Lee et al. (2024), Mirzadeh et al. (2023), Liu et al. (2023) Zhang et al. (2024)). Utilizing these adapters leads to inference speedups, as the amortized time complexity of an MLP layer reduces from $\mathcal{O}(d_{in} \cdot d_{hidden})$ to $\mathcal{O}(d_{in} \cdot d_{adapted})$, where $d_{in}$ is the input dimension of the MLP, $d_{hidden}$ is the hidden dimension, and $d_{adapted}$ is the average number of active neurons. For modern Transformer architectures, this results in practical speedups at the expense of negligible model quality decrease for a sufficient amount of average active neurons.

Unfortunately, adaptive compute methods using neuron adapters suffer from rapid performance degradation in modern Transformer architectures (Figs. 1a, 1c). We observe this issue arises from the limitations of the neuron-adaptive framework, which enforces dynamic neuron allocation based on activation values. Concretely, we believe this decline is attributed to this framework's lack of generalization across different layer types, reliance on sparse activation functions, and costly neuron masking techniques. First, neuron adapters, often tailored to MLPs, can not be directly applied to non-MLP layers, such as the Query, Key, and Value modules of attention layers, which lack neurons. In addition, these methods frequently depend on activation-induced sparsity, like that attained by ReLU (Agarap (2019)), making them ineffective with non-sparse activation functions like SwiGLU (Shazeer (2020), Zhang et al. (2024)). Further, while recent approaches like CATS (Lee et al. (2024)) or ReLUfication (Mirzadeh et al. (2023)) attempt to handle non-sparse activations, they inefficiently

---

[*]Corresponding author: `robgarct@stanford.edu`

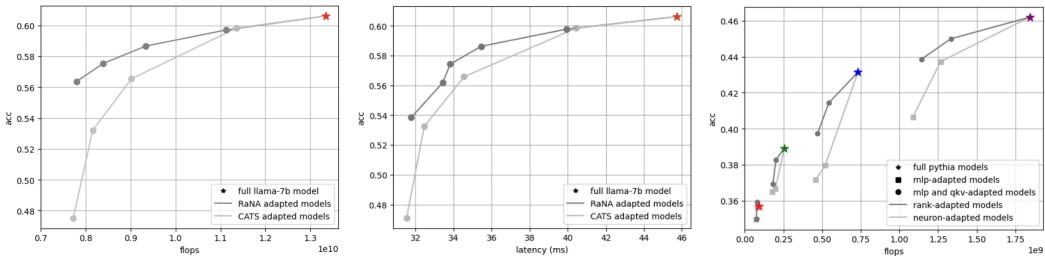

(a) Llama2-7b Accuracy v.s. FLOPs.  (b) Llama2-7b Accuracy v.s. Latency.  (c) Pythia Suite Accuracy v.s. FLOPs.

Figure 1: **RaNA improves accuracy-compute tradeoff over neuron adapters**. $y$-axis shows accuracy averaged over multiple downstream tasks (Sect. 5.1); for Figs. 1a and 1c, $x$-axis shows average FLOPs for a forward pass with sequence length 512; for Fig. 1b $x$-axis shows average per-token decoding latency over a sequence of 492 tokens with initial context lengths ranging from 1 to 1000. We compare RaNA-adapted models to neuron-adapted versions at various compression rates for (left) Llama2-7b and (right) Pythia models. Notably, RaNA accuracies decay slower as compression rates increase compared to neuron-adapters.

compute neuron-activations before determining which neurons to exclude. This is concerning as Transformers architectures like Llama (Touvron et al. (2023)), Gemma (Team et al. (2024)) or Mistral (Jiang et al. (2023)) rely on non-sparse activation functions, which makes sparsity-based neuron adapters ineffective for them, pushing us to rely on more computationally intensive neuron adapters.

To address these gaps in the neuron-adaptive setup, we propose the Adaptive Rank Allocation framework, together with the Rank and Neuron Allocator (RaNA) adapter. We note that this rank-adaptive framework is a more powerful generalization of the neuron-adaptive one. Concretely, we devise this framework from the observation that any linear layer can be decomposed into the product of two low-rank matrices and an adaptive router/masker, namely $\text{Linear}(x) = Wx \approx A(r(x) \odot Bx)$. Moreover, we develop RaNA in this framework, leveraging specific $A$ and $B$ decompositions and a masker $m(x)$ for Transformer layers, like QKV or MLPs. Notably, unlike neuron adapters, RaNA can be directly applied to any linear layer without relying on activation function sparsity. Further, when used with modern MLP layers with SwiGLU activations, it enables better distribution of compute across the MLP's linear layers.

We empirically validate the rank-adaptive framework and the RaNA adapter. We demonstrate that, similar to neuron-adaptive setups, the ranks of the $AB$ matrix decomposition in the proposed RaNA adapters have sparse importances, depending on the input (Figs. 2a, 2b), allowing us to dynamically prune them. We also show that RaNA adapters attain the lowest error in MLP layers when recovering the full MLP outputs, outperforming neuron-adaptive methods by 6.7, 18.1 and 7.4 percentage-points on Llama2-7b, Gemma-2b and Pythia-160M respectively. Further, we show the effectiveness of RaNA in modern Transformer architectures by applying it to Llama (Touvron et al. (2023)) and Gemma (Team et al. (2024)). Notably, RaNA closes the gap between full-model and adapted-model performance, as it outperforms the state-of-the-art method CATS (Lee et al. (2024)) on multiple compression rates (Tabs. 1, 2). Concretely, at a ∼44% FLOP compression rate, RaNA achieves an average improvement of 4 perplexity-points and 8 percentage-points in benchmark accuracy for Llama2-7b, while for Gemma-2b, it improves perplexity by 7 points and accuracy by 5 percentage-points, compared to prior adapters. Finally, to assess the effectiveness of RaNA's applicability to different types of layers and activations, we apply it to the Pythia suite (Biderman et al. (2023)), a set of varied-sized GPTNeox (Black et al. (2022)) models. Here we also observe that, when comparing their performance to conventional neuron-adaptive methods, it consistently attains better perplexity and downstream task performance (Figs. 1c, 4). Further, code is available at https://github.com/Roberto09/RaNA.

## 2 RELATED WORK AND PRELIMINARIES

Here we discuss key work in neuron-adaptive methods and examine neuron adapters for Transformers with sparse activations and with modern non-sparse activations.

**Neuron-adaptive framework**: Adaptive compute methods (Han et al. (2021)) are a popular approach to speed up inference in Transformer (Vaswani et al. (2023)) architectures. A concrete instance of them are neuron adapters, which dynamically allocate neurons of the MLP layers to different inputs using a masker. These neuron-adaptive methods for Transformers stem from the observation that neuron importance is sparse, and while Transformers allocate equal resources to all tokens, some tokens require less compute based on context (Li et al. (2023b)). Empirically, redundant computations are common due to model over-parameterization (Frankle & Carbin (2019)).

Nevertheless, the neuron-adaptive framework is constraining, as it only allows us to dynamically allocate neurons, which are only present on MLP layers of Transformers. Additionally, neuron adapters do not generalize well to different activation functions in the MLP layers (Zhang et al. (2024)). Hence, previous work has focused on creating adapters specific to certain activation functions, namely sparse activation functions and non-sparse activation functions.

**Neuron adapters for sparse activations**: In the case of Transformers leveraging sparse activation functions like ReLU (Agarap (2019)), neuron adapters rely on the abundance of 0-valued neuron activations (Li et al. (2023a)). These adapters commonly work by using a small MLP masker that predicts whether a neuron is going to be active for a given input, then only computing that neuron if its prediction is positive. Concretely, consider a conventional MLP layer with a ReLU activation:

$$\text{MLP}_{ReLU}(x) = W_{down}(\text{ReLU}(W_{up}x))) \tag{1}$$

where $W_{up} \in \mathbb{R}^{d,h}$ and $W_{down} \in \mathbb{R}^{h,d}$ are the Up-Projection and Down-Projection matrices of the MLP layer. Then, the neuron adapted version of this MLP follows:

$$\text{MLP}'_{ReLU}(x) = W_{down}(m(x) \odot \text{ReLU}(W_{up}x)) \tag{2}$$

where $m(x) : \mathbb{R} \to \{0,1\}^h$ is the binary masker, which often is parameterized by a small MLP. While effective in these type of models, given their strong reliance on the sparsity of neuron activations, this adapters have unfortunately shown poor performance when applied to Transformers with non-sparse activations (Zhang et al. (2024)).

**Neuron adapters for non-sparse activations**: Modern Transformer architectures, like those used in popular models like Llama (Touvron et al. (2023)), Gemma (Team et al. (2024)), Mistral (Jiang et al. (2023)) or GPTNeox (Black et al. (2022)), leverage non sparse activation functions like GeLU (Hendrycks & Gimpel (2023)) for GPTNeox and the more popular SwiGLU (Shazeer (2020)) for the others. Concretely, an MLP layer with a SwiGLU activation follows:

$$\text{MLP}_{SwiGLU}(x) = W_{down}(\text{SiLU}(W_{gate}x) \odot W_{up}x) \tag{3}$$

where $W_{up} \in \mathbb{R}^{d,h}$, $W_{down} \in \mathbb{R}^{h,d}$ and $W_{gate} \in \mathbb{R}^{d,h}$ are the Up-Projection, Down-Projection and Gate-Projection matrices of the MLP layer.

Recent work has sought to improve the performance of conventional neuron adapters on non-sparse activations by developing methods specifically tailored towards them, such as CATS (Lee et al. (2024)) and ReLUfication (Mirzadeh et al. (2023)). While these approaches have demonstrated good performance in SwiGLU-based Transformers, they suffer from inefficiencies. The inefficiency stems from their reliance on calculating exact activation values prior to applying thresholding. For instance, CATS adapters compute the entire output of the Gate-Projection layer before selecting which neurons to retain. At large compression rates, this enforces a sub-optimal FLOP allocation imbalance across the Up, Down and Gate projection layers of the MLP, where the Gate-Projection layer receives most of the FLOPs.

## 3 ADAPTIVE RANK ALLOCATION FRAMEWORK

We begin by presenting the Adaptive Rank Allocation Framework, which addresses the shortcomings of the neuron-adaptive approach, specifically its limitation to neuron adaptation and its frequent reliance on neuron sparsity. Adaptive compute frameworks generally involve two key parts: dynamically allocated components and a function to determine their use based on the input. We make the observation that such an adaptive compute setup can be applied at the granularity of linear layers, where the dynamically allocated components are the ranks of a weight matrix, while the function that determines which ranks are leveraged by a given input is a router or masker. This is the main observation that motivates our Adaptive Rank Allocation Framework, which can be applied to arbitrary linear layers without relying on activation function sparsity.

Concretely, consider any linear layer $\text{Linear}(x) = Wx$, where $W \in \mathbb{R}^{o,i}$. Under the Adaptive Rank Allocation framework, we replace this linear layer by:

$$\text{Linear'}(x) = A(r(x) \odot Bx) \tag{4}$$

This simple setup has two parts. First, we have a set of static matrices A and B, where $A \in \mathbb{R}^{o,d}$ and $B \in \mathbb{R}^{d,i}$. Second, we have the adaptive component $r(x)$, where $r : \mathbb{R}^i \to \mathbb{R}^d$. Ideally we want $r(x)$ to be sparse, as that allows us to save I/O and floating-point operations. If $r(x)$ is sparse, our rank adapted layer from Eqn. 4 becomes essentially a low-rank matrix multiplication, where the the low-rank matrix $(A \text{ diag}(r(x)) \ B)$ has rank $= \|r(x)\|_0$. Notably, the FLOPs consumed by such a matrix multiplication are proportional to the rank of such matrix, hence why sparsity is desirable.

**Generalization of Neuron-Adaptive Methods**: We note that the Adaptive Rank Allocation framework is a strict generalization of the neuron-adaptive one, where neuron adapters can be viewed as a specific instance of rank adapters with appropriate choices of $A$, $B$ and $r(x)$. In Prop. 1, we illustrate this for ReLU-based MLPs, but we note the proof easily extends to other activation functions:

**Proposition 1.** *Consider an $MLP(x)_{ReLU}$ layer (Eqn. 1) and its neuron adapted version $MLP'(x)_{ReLU}$ (Eqn. 2). Then, there exists a rank adapted $MLP^*_{ReLU}$ (i.e. an MLP whose linear layers have been rank adapted) s.t. $MLP^*_{ReLU}(x) = MLP'(x)_{ReLU}$ for all $x \in \mathbb{R}^i$.*

*Proof.* Let $\text{MLP}^*_{ReLU}(x) = A_{down}(r_{down}(x) \odot B_{down}(\text{ReLU}(A_{up}(r_{up}(x) \odot B_{up}x))))$.

Then, by setting $A_{down} := W_{down}$, $B_{down} := \mathbf{I}$, $A_{up} := \mathbf{I}$, $B_{up} := W_up$, $r_{down} := m_{down}$, and $r_{up} := \mathbf{1}$, it follows that:

$$\text{MLP}^*_{ReLU}(x) = W_{down}(m_2(x) \odot \mathbf{I}(\text{ReLU}(\mathbf{I}(\mathbf{1} \odot W_{up}x))))$$
$$= W_{down}(m_2(x) \odot \text{ReLU}(W_{up}x)) = \text{MLP}'_{ReLU}(x).$$

In summary, this result shows that rank adaptation generalizes neuron adaptation, highlighting the increased versatility and applicability of the Adaptive Rank Allocation framework.

## 4    RANK AND NEURON ALLOCATOR ADAPTERS

Here, we introduce Rank and Neuron Allocator (RaNA) adapters as instances of the Adaptive Rank Allocation framework. RaNA adapters leverage Linear Layer Rank adapters (Sect. 4.1) at their core, which operate at the granularity of linear layers. Such Linear Layer Rank adapters employ approximately optimal $A$ and $B$ matrices in tandem with a dynamic rank allocation mechanism based on the input distribution. Notably, while Linear Layer Rank adapters are effective in QKV, Up-Projection, and Gate-Projection layers, they are inefficient in Down-Projection layers when used as stand-alone adapters. To overcome this limitation, we propose RaNA adapters.

RaNA adapters operate at the level of QKV and MLP layers, combining Linear Layer Rank adapters with neuron-thresholding techniques, a setup that has demonstrated the best empirical performance when adaptively compressing layers (Figs. 3a, 3b, 3c, 3d). Notably, these adapters address the inefficiencies of previous non-sparse neuron adapters by avoiding the costly exact computation of specific linear layer outputs prior to applying neuron thresholding. We empirically demonstrate their effectiveness in model accuracy and perplexity compared to other adapters (Tabs. 1, 2).

### 4.1    LINEAR LAYER RANK ADAPTERS

The Adaptive Rank Allocation framework, unlike the neuron-adaptive one, does not impose limitations on the $A$ or $B$ matrices or the routing function $r(x)$ when adapting a linear layer through $\text{Linear}'(x) = A(m(x) \odot Bx) \approx Wx$ . We leverage these properties to introduce the Linear Layer Rank Adapters. Here, we want to find the optimal $A$, $B$ and $r(x)$ that, in expectation over the input distribution, best recover the original output of the linear layer. Formally, we want:

$$\text{argmin}_{A,B,r} E_x(\|Wx - A(r(x) \odot Bx)\|_F^2) \tag{5}$$

Evidently, optimally finding $A$, $B$ and $r$ is non-trivial. Hence, we choose to address this problem by breaking it into two steps. First, we propose A and B matrices that solve a similar problem to Eqn. 5, which ignores the routing function. Then, with our election of A and B matrices, we propose an optimal input-dependent masker, which serves as a router, for our Linear Layer Rank Adapter.

**A and B matrices**: First, we devise a set of A and B matrices for our rank-adapted layer Linear'$(x) = A(r(x) \odot Bx)$. We start off by relaxing the optimization problem from Eqn. 5 . Concretely, we remove the router from the picture and frame the problem as finding fixed low-rank matrices $A_r$ and $B_r$ of rank $r$ that minimize our objective. Namely, the relaxed objective becomes:

$$\text{argmin}_{A_r, B_r} E_x(\|Wx - A_r B_r x\|_F^2) \tag{6}$$

Note that this relaxed problem is not only convenient, but also a decent choice, as it is equivalent to solving the original problem from Eqn. 5 with a fixed router $r(x)$ that always outputs 1's for the first $r$ elements of the output vector and 0's for the rest. In practice, we do not have access to the actual distribution of the inputs $x$, hence we reframe the problem to its empirical version:

$$\text{argmin}_{A_r, B_r} \|WX - A_r B_r X\|_F^2 \tag{7}$$

where each column of $X \in \mathbb{R}^{i,k}$, namely $x_i$, is a hidden-state input that the linear layer from our pretrained model observes in practice. Notably, we used $k = 32,000$ samples in our experiments.

We can then analytically find the optimal $A_r$ and $B_r$ matrices that solve this problem.

**Theorem 1.** *Let $U_r \in \mathbb{R}^{o,r}$ be the first $r$ singular vectors of $WX$, then by the Eckart-Young theorem (Eckart & Young (1936)), the $A_r$ and $B_r$ that optimize the objective from Eqn. 7 are:*

$$A_r := U_r, B_r := U_r^T W.$$

The proof for Theorem 1 is provided in Appendix A.1. Therefore, we pick our $A,B$ matrices as $A := U$, $B := U^T W$. Next, we demonstrate how to pick a masker/router to effectively leverage these matrices in an input-adaptive manner.

**B-Masker**: With our $A = U$ and $B = U^T W$ matrices at hand, we devise a sparse router $r(x)$ function that minimizes the adapter's error (Eqn. 5). For simplicity, we opt for a binary $r(x)$, i.e. a masker $r(x) = m(x) : \mathbb{R}^i \to \{0,1\}^i$ that allows us to allocate different ranks (and hence different amount of FLOPs) to different input hidden states. Formally, we want an $m(x)$ that optimizes:

$$\text{argmin}_{m(x)} \sum_i^k \|Wx_i - U(m(x_i) \odot U^T W x_i)\|_F^2, \quad \text{s.t. } \mathbb{E}_x[\|m(x)\|_0] = r$$
$$\equiv \text{argmax}_{m(x)} \sum_i^k \|m(x_i) \odot U^T W x_i\|_F^2, \quad \text{s.t. } \mathbb{E}_x[\|m(x)\|_0] = r \tag{8}$$

The constraint in Eqn. 8 enforces that expected rank of the matrix $(A \, \text{diag}(m(x)) \, B)$ is $r$. This constraint directly controls the FLOPs that the rank-adaptive model will consume on average.

Observe that, for any hidden state $x$, we can compute the contribution of each rank in our adapter to the output of the original linear layer. To do this, we examine the contribution of each column vector in $A = U$ (equivalently, the contribution of each rank of $A \, B$) to the output $o = \text{Linear}(x) \approx A(m(x) \odot Bx)$. Specifically, we note that the contribution to the Frobenius norm $\|o\|_F^2$ from the $i$-th column vector of $A$, $u_i$, is given by $(Bx)_i^2$, due to the orthogonality of the columns in $A = U$. This allows us to identify the important ranks of our $A$, $B$ decomposition for a given input $x$ and create a sparse masker $m(x)$ for it (Figs. 2a, 2b), which just keeps the most descriptive ranks. We call this the B-Masker, as it uses the B matrix to select the most important ranks for $x$. Formally:

$$m(x)_i = \text{B-masker}(x)_i = \mathbf{1}\{(Bx)_i^2 \geq t\} \tag{9}$$

where $\mathbf{1}\{\cdot\}$ is the indicator function, and the threshold $t$ is picked so that, on average, a desired amount of FLOPs is consumed by our adapted layer. We note that the B-masker is efficient to use when the matrix from our linear layer $W \in \mathbb{R}^{(o,i)}$ has an output dimension $o$ that is bigger than its inner dimension $i$, as this makes computing $(Bx)^2$ cheap. Hence, it is convenient to use this masker in practice for the Up-Projection, Gate-Projection and QKV layers of our models.

**MLP-Sigmoid Masker**: An alternative option to the B-masker, which has the potential to be more efficient, and that works for any linear layer, is a small MLP masker with a Sigmoid activation function, just like that used commonly in the neuron-adaptive literature (Liu et al. (2023), Zhang et al. (2024)). Concretely, we use the parametrization $m(x) = \sigma(CDx)$ where $C \in \mathbb{R}^{r,r'}$, $D \in \mathbb{R}^{r',i}$, $i$ is the dimension of the hidden states $x$, $r'$ is the inner dimension of the predictive masker, and $r$ is the rank of the matrix $W$ from the linear layer the masker corresponds to. In our experiments, we train this masker on a binary cross-entropy loss to match the output of the B-masker.

## 4.2 FUSING RANK ADAPTERS AND NEURON THRESHOLDING FOR RaNA

Building on the insights from the previous section, we observe that Linear Layer Rank Adapters independently perform well for QKV, Up-Projection, and Gate-Projection layers due to their use of tall and narrow matrices. However, they may encounter difficulties with Down-Projection layers, which involve short and wide matrices. To overcome this limitation, RaNA combines Linear Layer Rank Adapters with neuron-thresholding adapters, specifically for the Down-Projection layers. Further, RaNA implements a FLOP allocation procedure across the various components of the adapter, addressing the imbalance found in previous neuron-adaptive methods, where FLOP distribution is heavily skewed toward specific components.

**RaNA in QKV layers**: For the QKV layers, we just replace the linear QKV layers with Linear Layer Rank Adapters, namely:

$$\text{QKV}(x) = Wx \approx \text{QKV}'(x) = A(m(x) \odot Bx) \tag{10}$$

where $m(x)$ is a B-masker, as we find it outperforms the MLP-Sigmoid masker (Fig. 3d).

**RaNA in MLP layers**: For MLP layers, we demonstrate how RaNA is applied to SwiGLU-based MLPs and note that its application to MLPs with other activation functions follows the same approach, excluding the Gate-Projection adapter. Namely, we use Linear Layer Rank Adapters for the Up-Projection and Gate-Projection matrices and use neuron-thresholding for the Down-Projection matrices. Formally, our RaNA adapted MLP'$(x)$ is described by the following:

$$
\begin{aligned}
\text{MLP}'(x) &= \text{Down}'(\text{SiLU}(\text{Gate}'(x)) \odot \text{Up}'(x))) \\
\text{where} \quad \text{Up}'(x) &= A_{up}(m_{up}(x) \odot B_{up}x), \\
\text{Gate}'(x) &= A_{gate}(m_{gate}(x) \odot B_{gate}x), \\
\text{Down}'(x) &= W_{down}(m_{down}(x) \odot x)
\end{aligned}
\tag{11}
$$

where $m_{up}(x)$ and $m_{gate}(x)$ are B-maskers and $m_{down}(x)$ is a simple neuron thresholding masker:

$$m(x)_i = \text{neuron-thresholding}(x)_i = \mathbf{1}\{|x_i| \odot ||W_{i,:}^{down}||_F \geq t\} \tag{12}$$

**RaNA FLOP Allocation**: Prior neuron-adapters for non-sparse activation functions are forced to ineffectively allocate FLOPs across their components (Sect. 2). On the other hand, RaNA's flexibility permits us to freely distribute FLOPs across the adapter components, therefore we propose a FLOP allocation strategy specifically for RaNA. At the Linear Layer Rank Adapter level, we perform a simple line search to balance FLOPs between the B-Masker and the target sparsity, selecting the configuration that minimizes the output error with respect to the original linear layer. Similarly, at the MLP level, we conduct a grid search to distribute FLOPs among the Up', Gate', and Down' layers, with each component further balancing FLOPs between its masker and target sparsity. We then retain the configuration that achieves the greatest error reduction in the MLP output. Please refer to Appendix A.4 for a pseudocode implementation of RaNA.

## 5 EXPERIMENTS AND RESULTS

In this section, we present experiments and results for the following claims:

1. **The contribution of ranks in Linear Layer Rank Adapters is sparse**. Just like this is a desirable property for neuron-adaptive approaches to work well, it is also one for the Linear Layer Rank Adapters, which we use in RaNA. Notably, the distribution of the contributions of each rank in the $AB$ decomposition (to the recovery of the original output of the linear layer) of our rank adapters is concentrated at 0, and is heavy tailed (Figs. 2a, 2b). This validates Linear Layer Rank adapters and practically means that we can effectively mask out many of the ranks with near-0 contributions.

2. **RaNA adapters attain lowest errors on Transformer layers when reconstructing original layer outputs**. Concretely, we conduct a study of the reconstruction error in the layer outputs of the Llama2-7b, Gemma-2b and Pythia-160M models using multiple compression methods, targeting a ~50% reduction in the FLOPs of the compressed layers. When compared to other adapters, RaNA attains the lowest mean squared reconstruction error of the outputs of the original layers (Figs. 3a, 3b, 3c, 3d). Notably, RaNA achieves average errors of 10.5% and 2.18%, surpassing the 17.22% and 20.31% attained by CATS on MLP layers for Llama2-7b and Gemma-2b respectively. Likewise, in Pythia-160M, RaNA

achieves average errors of 7.87% and 0.36% in MLP and QKV layers respectively, outperforming the 15.23% and 1.29% errors obtained with neuron adapters and SVD-based adapters.

3. **RaNA outperforms neuron adapters in perplexity and downstream tasks**. Using RaNA, we attain lower perplexity and higher average accuracy on multiple NLP benchmarks in modern Transformers compared to neuron-adapters. Notably, RaNA outperforms previous neuron adapters across multiple FLOP compression levels (Tabs. 1, 2, Fig. 1a) for SwiGLU based architectures. In the case of Llama2-7b, we achieve an improvement of 8 percentage-points in accuracy and 4 perplexity-points on a 42% model compression rate. Similarly, for Gemma-2b, we attain an improvement of 5 percentage-points in accuracy and 7 perplexity-points on a 46% model compression rate. We also show that RaNA not only preserves a theoretical advantage in the accuracy-FLOP trade-off but also practically in the accuracy-latency trade-off (Fig. 1b). Further, RaNA outperforms conventional neuron-adapters in GeLU-based Pythia models across multiple model sizes and compression levels (Figs. 1c, 4).

## 5.1 EXPERIMENTAL SETUP

Here we describe the settings under which we run our experiments.

**Models and Adapters:** We conduct experiments using Llama2-7b, Gemma-2b, and multiple GPT-Neox models from the Pythia suite. Notably, we fine-tuned Gemma-2b on the RedPajama dataset for ∼42M tokens before applying adapters to it, so as to align it with the data used in subsequent adaptation and fine-tuning steps. In Sect. 5.3, we evaluate the following adapters, which are mainly implemented in PyTorch.

- *RaNA*: Our proposed adapter from Sect. 4.2.
- *CATS*: A state-of-the-art adapter for MLP layers leveraging SwiGLU activations. We refer to their work (Lee et al. (2024)) for more details.
- *SliceGPT*: A recent structured pruning method, which compresses linear layers by rotating and slicing them, including MLP and QKV layers. We refer to their work (Ashkboos et al. (2024)) for more details.
- *Neuron-Adaptive*: A standard neuron adapter for MLP layers with a small MLP masker, such as that leveraged by Zhang et al. (2024) or Liu et al. (2023), using 6% of MLP FLOPs for the masker, as done by Zhang et al. (2024).
- *Linear-Layer-Rank-Adapters with MLP maskers (LLRA)*: An adapter that applies Linear Layer Rank adapters leveraging an MLP-based masker (Sect. 4.1) to all linear layers in QKV and MLPs.

For output error assessments in Sect. 5.3, we additionally use a fixed low-rank singular value decomposition (SVD) for comparison.

**Datasets:** We use the RedPajama (Computer (2023)) dataset for Llama2-7b and Gemma-2b, and the Pile (Gao et al. (2020)) dataset for Pythia models when evaluating rank contribution sparsity (Sect. 5.2), output errors (Sect. 5.3), and perplexity (Sect. 5.3), and for devising any data-dependent adapter component (e.g. A and B matrices in RaNA, the activation threshold in CATS and the slicing and rotating procedure of SliceGPT). Downstream-task performance is assessed using HellaSwag (Zellers et al. (2019)), PIQA (Bisk et al. (2019)), WinoGrande (Sakaguchi et al. (2019)), Arc-Easy (Clark et al. (2018)), Arc-Challenge (Clark et al. (2018)) and RACE (Lai et al. (2017)).

**Fine-tuning:** To assess accuracy and perplexity (Sect. 5.3), we fine-tune adapted models using the Huggingface library (Wolf et al. (2020)) and LoRA adapters (Hu et al. (2021)) for ∼31M tokens on Llama2-7b and Gemma-2b, with an AdamW optimizer, where learning rates were determined from various options, depending on the model's performance following ∼6M tokens of training. In a similar setup, Pythia models are fine-tuned for ∼61M tokens with the exception of not leveraging LoRA adapters.

**Performance Evaluations:** We use LM-evaluation harnesses (Gao et al. (2023)) for downstream-task performance evaluation in a zero-shot setting for Llama2-7b and Pythia models, and a five-shot

setting for Gemma-2b. Perplexity is measured on a held-out subset of each model's fine-tuning dataset. Further, FLOP compression is assessed by measuring the average FLOPs required to decode 512-token sequences.

**Latency Evaluations:** For our latency evaluations, we leverage 100 sequences from the RedPajama dataset, where adapted models are timed in the task of decoding a sequence of 492 tokens with an initial context ranging from 1 to 1000 tokens. Evaluations are performed on an NVIDIA L40S GPU.

## 5.2 RANK CONTRIBUTION SPARSITY

A key property of adaptive compute methods is having sparse contributions from the pruned components to the module's output. In RaNA, we adaptively prune the ranks of the $AB$ matrix in the decomposition $Wx \approx A(m(x) \odot Bx)$, which effectively means pruning the column vectors of the $A$ matrix (Sect. 4.1). Concretely, we aim for sparsity in the contribution of each column vector of the $A$ matrix to the output of the linear layer, which we obtain by measuring $(Bx)_i^2$. To measure this, we study the histograms of these contributions in Llama2-7b, Gemma-2b, and Pythia-160M models, shown in Figs. 2a, 2b. The distributions exhibit heavy tails with concentrations near zero, allowing us to mask out irrelevant values and retain the most impactful ones.

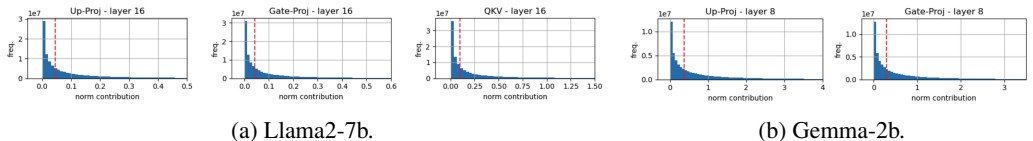

(a) Llama2-7b.                                                  (b) Gemma-2b.

Figure 2: **The contribution of ranks in Linear Layer Rank Adapters is sparse for multiple layer types** (Sect. 5.2). Histograms outline the contribution of different column-vectors from the $A$ matrix in the Linear Layer Rank Adapter decomposition $Wx \approx A(m(x) \odot Bx)$ to the original layers for Llama2-7b (left) and Gemma-2b (right). Red dashed line indicates a 50% sparsity threshold.

## 5.3 RaNA EVALUATIONS

**Output Errors**: To assess the compression capabilities of RaNA, we apply it to the MLP and QKV layers of Llama2-7b, Gemma-2b and Pythia-160M. Further, we measure the error that RaNA and other adapters induce in these layers when adapting them to consume $\sim$50% of their FLOPs, as it is a common compression ratio in the pruning literature (Ma et al. (2023), Ashkboos et al. (2024)). For the MLPs, we measure the normalized error $\frac{|\text{MLP}(x)-\text{MLP}'(x)|_2^2}{|\text{MLP}(x)|_2^2}$, where MLP' is the adapted MLP; we do the analogous measurement for the QKV layers too. Intuitively, an effective adapter produces small errors, as that allows it to recover the model's original behavior better. Notably, from Figs. 3a, 3b, 3c, 3d, we can observe that RaNA attains the lowest error across all layers when compared to neuron-adapters and other adapter types. Concretely, in the case of Llama2-7b, RaNA attains an average error of 10.5%, while CATS attains an average error of 17.22% in MLP layers. Further, for Gemma-2b's MLPs, RaNA achieves an error of 2.18%, while CATS attains an error of 20.31%. Similarly, for Pythia-160M, RaNA achieves average errors of 7.87% and 0.36% across MLP and QKV layers respectively, while other approaches attain average errors of 15.23% and 0.36%. This demonstrates RaNA's capacity to effectively compress modern Transformer layers, which we further show translates into practical downstream task performance and perplexity improvements.

**Downstream Task Performance and Perplexity**: To examine the performance of RaNA adapters beyond their compression capabilities, we measure the perplexities and downstream task performance when applied to Llama2-7b, Gemma-2b, and a set of varied sized Pythia models. For these evaluations, we first apply the given adapter to the MLP and/or QKV layers of the model at hand, targeting a specific FLOP reduction ratio. Concretely, for Llama and Pythia models we apply RaNA to MLP and QKV layers, while for Gemma we only apply it to MLP layers. Notably, we opted for not adapting QKV layers in Gemma for simplicity, as they constitute just a small proportion of FLOPs ($\sim$5%) relative to the MLP layers. Further, we fine-tune the adapted models. Finally, we measure their perplexity on a held-out subset of the dataset used for fine-tuning and measure their accuracies for multiple NLP benchmarks.

From Tabs. 1, 2, we observe that RaNA outperforms the state-of-the-art CATS adapters across a varied set of compression rates in SwiGLU based Transformers. Notably, not only does it outper-

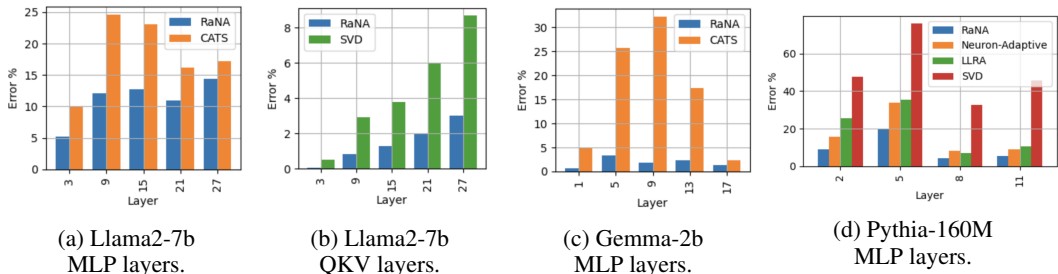

(a) Llama2-7b MLP layers.    (b) Llama2-7b QKV layers.    (c) Gemma-2b MLP layers.    (d) Pythia-160M MLP layers.

Figure 3: **RaNA adapters attain lowest errors on Transformer layers when reconstructing original layer outputs.** $y$-axis shows the error percentage; $x$-axis shows layer number. Errors induced by different adapters are compared when compressing layers of Llama2-7b, Gemma-2b and Pythia-160M to 50% FLOPs. RaNA attains the lowest error consistently across model layers (Sect. 5.3).

| Method | FLOP Compression Rate | Arc Easy | Arc Challenge | HellaSwag | PIQA | RACE | WinoGrande | Avg Acc | PPL |
|---|---|---|---|---|---|---|---|---|---|
| Llama2-7b | 0% | 76.26% | 43.52% | 57.08% | 78.07% | 39.62% | 69.14% | 60.61% | 6.39 |
| RaNA | 42% | 68.81% | 36.86% | 51.96% | 74.76% | 39.71% | 66.14% | **56.37%** | **8.04** |
| CATS | 42% | 56.31% | 27.13% | 41.01% | 68.28% | 35.12% | 57.22% | 47.51% | 12.36 |
| SliceGPT | 42% | 44.91% | 21.84% | 37.92% | 61.64% | 37.22% | 55.72% | 43.21% | 13.69 |
| RaNA | 30% | 72.85% | 39.76% | 54.63% | 77.42% | 40.48% | 66.85% | **58.67%** | **7.29** |
| CATS | 32% | 69.49% | 36.01% | 53.16% | 76.28% | 38.76% | 65.59% | 56.55% | 7.55 |
| SliceGPT | 31% | 52.36% | 26.62% | 42.83% | 67.74% | 38.28% | 60.30% | 48.02% | 11.23 |
| RaNA | 17% | 73.86% | 41.55% | 56.09% | 77.80% | 39.52% | 69.38% | 59.70% | 6.63 |
| CATS | 15% | 75.17% | 41.55% | 56.96% | 77.48% | 39.52% | 68.27% | **59.82%** | **6.37** |
| SliceGPT | 17% | 58.96% | 31.40% | 48.40% | 72.52% | 39.62% | 63.38% | 52.38% | 9.22 |

Table 1: **RaNA outperforms neuron-adapters in perplexity and accuracy in Llama2-7b**. Perplexity is measured on ∼300K tokens of RedPajama. Average accuracy is aggregated over the listed benchmarks. The compression rate outlines the average FLOP compression rate for decoding a 512-token long sequence.

forms when applied to MLP and QKV layers, as shown for Llama2-7b (Tab. 1), but it also does when only applied to the MLP layers as shown for Gemma-2b (Tab. 2). Particularly, for Llama2-7b, RaNA improves over neuron-adaptive methods by attaining 4 less perplexity-points and 8 more percentage-points in accuracy when reducing FLOPs by 42% on the overall model. Moreover, both RaNA and CATS outperform SliceGPT in both perplexity and downstream task performance (Tab. 1, Fig. 5) for all FLOP compression rates, highlighting the benefits of adaptive compression methods compared to static ones. Similarly, at a 46% compression, for Gemma-2b, RaNA achieves 7 less perplexity-points and 5 more percentage-points in accuracy than prior neuron adapters.

We attribute RaNA's strong performance to its notable compression capacity, its ability to more evenly distribute FLOPs across the Up-Project, Down-Project, and Gate-Project layers, and its direct applicability to QKV and MLP layers, all of which posed challenges for previous neuron adapters. In addition, from Figs. 1c and 4, we observe that RaNA applied to MLP and QKV layers outperforms neuron adapters across the varied sized set of GeLU based Pythia models in both average accuracy and perplexity. This highlights RaNA's general applicability to modern Transformers with non-sparse activations, such as SwiGLU in Llama and Gemma or GeLU in Pythia. Further, these results show RaNA's effectiveness beyond individual layer compression, demonstrating practical improvements over neuron adaptive techniques in perplexity and downstream task evaluations. Moreover, we attribute RaNA's and CATS' strong performance over SliceGPT, a static structured pruning method, to their adaptive nature. Different from static approaches, RaNA and CATS dynamically adjust weight matrices based on the input, enabling a more efficient performance-compute trade-off in practice.

| Method | FLOP Compression Rate | Arc Easy | Arc Challenge | HellaSwag | PIQA | RACE | WinoGrande | Avg Acc | PPL |
|--------|------|------|------|------|------|------|------|------|------|
| Gemma-2b | 0% | 59.47% | 29.95% | 46.90% | 70.24% | 36.17% | 56.83% | 49.93% | 11.05 |
| RaNA | 44% | 58.84% | 29.18% | 42.34% | 69.31% | 34.83% | 54.46% | **48.16%** | **13.83** |
| CATS | 47% | 51.43% | 24.57% | 35.07% | 65.94% | 28.42% | 52.64% | 43.01% | 21.02 |
| RaNA | 32% | 61.74% | 31.66% | 45.38% | 70.40% | 35.89% | 51.62% | **49.45%** | **11.74** |
| CATS | 34% | 59.55% | 29.61% | 44.51% | 70.95% | 34.55% | 54.38% | 48.92% | 12.60 |
| RaNA | 19% | 59.26% | 29.78% | 46.42% | 69.31% | 35.98% | 55.01% | 49.29% | **11.18** |
| CATS | 20% | 62.79% | 32.94% | 47.84% | 72.25% | 35.50% | 55.41% | **51.12%** | 11.41 |

Table 2: **RaNA outperforms neuron-adapters in perplexity and accuracy in Gemma-2b**. Perplexity is measured on ∼300K tokens of RedPajama. Average accuracy is aggregated over the listed benchmarks. The compression rate outlines the average FLOP compression rate for decoding a 512-token long sequence.

**Latency Evaluations**: From Fig.1a, RaNA presents an improved theoretical accuracy-FLOPs trade-off over previous adaptive methods. To validate its practical benefits, we evaluate latency by constructing a accuracy-latency curve. Since a naive PyTorch implementation lacks practical speedup, we integrated a custom masked GEMV kernel to RaNA for both its linear-layer rank adapter and neuron adapter. Concretely, we leverage the masked GEMV kernel from CATS, implemented in Triton (Tillet et al. (2019)).

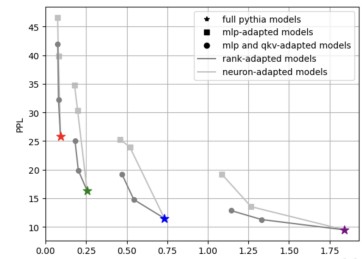

Leveraging a masked GEMV kernel in RaNA is possible as, in the linear-layer rank adapter, $A(m(x) \odot Bx)$ reduces to a masked matrix-vector multiplication, with $A$ as the matrix, $m(x)$ as the mask, and $Bx$ as the vector. Similarly, in the neuron adapter, $W(m(x) \odot x)$ performs a masked multiplication with $W$ as the matrix, $m(x)$ as the mask, and $x$ as the vector. Fig.1b highlights RaNA's capacity to realize practical speedups in various compression rates for Llama2-7b, positioning RaNA as a practical adapter for accelerating Transformer inference.

Figure 4: Perplexities are measured across adapted Pythia models, as a complementary measurement to accuracies outlined in Fig 1c. $y$-axis shows perplexity measured over ∼300K tokens of the Pile dataset (Sect. 5.1); $x$-axis shows average FLOPs for a forward pass with sequence length 512.

## 6 CONCLUSION AND FUTURE WORK

We present the Adaptive Rank Allocation framework and RaNA adapters, designed to address the limitations of neuron-adaptive methods in modern Transformer architectures. By moving beyond neuron-based adaptation, RaNA is effectively applicable to both MLP and QKV layers, leveraging low-rank matrix decompositions and adaptive routers. Empirical results demonstrate that RaNA achieves greater performance over existing neuron-adaptive methods like CATS, providing improvements in perplexity and accuracy across benchmarks when reducing FLOPs. For instance, RaNA yields 4 fewer perplexity-points and an improvement of 8 percentage-points in accuracy for Llama2-7b when compressing FLOPs by 42%.

Future work seems promising. First, exploring the applicability of RaNA to other architectures, such as vision transformers or those leveraging different activation functions than the ones studied in this work could extend its impact. Additionally, investigating alternative matrix decomposition techniques and router configurations within Linear Layer Rank adapters could further improve RaNA's effectiveness. Finally, exploring a FLOP allocation strategy at the model level, rather than focusing solely on individual layers, presents a promising opportunity to improve RaNA's overall capacity.

ACKNOWLEDGMENTS

JL is supported by the Department of Energy Computational Science Graduate Fellowship under Award Number DE-SC0023112.

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

# A APPENDIX

## A.1 PROOF OF THEOREM 1

Here we prove Theorem 1.

*Proof.* Let $M_r$ be a rank $r$ matrix. The matrix $M_r$ that obtains the lowest possible error ($\|WX - M_r\|_F^2$) is the rank-$r$ singular value decomposition of $WX$, namely $M_r = U_r \Sigma_r V_r^T$. Now, solving for $A_r$, $B_r$:

$$A_r B_r X = M_r = U_r \Sigma_r V_r^T \implies A_r B_r = U_r \Sigma_r V_r^T X^+$$

where $X^+$ is the pseudo-inverse of $X$. Hence, we have found the optimal $A_r B_r$ for eqn. 7, namely:

$$A_r B_r = U_r \Sigma_r V_r^T X^+ = U_r (U_r^T W).$$

## A.2 ABLATIONS AND ADDITIONAL RESULTS

**Compressing MLPs only vs. MLPs + QKVs**:

As an ablation study, we look at how compressing both MLPs and QKVs compare to only compressing MLPs to the same FLOP compression ratio in Llama2-7b. In addition, we study how our FLOP-allocation algorithm for MLP layers impacts performance by evaluating an adapted model that leverages the FLOP allocation procedure in the MLP layers and evaluating the same adapted model with the exception that it uniformly allocates FLOPs across the components of each individual MLP layer.

Concretely, we compress three Llama2-7b models by ∼31% of their FLOPs and evaluate their perplexity on ∼300K tokens of the RedPajama dataset, without fine-tuning. First, we look at a vanilla RaNA adapted model, which leverages the FLOP allocation algorithm at the MLP level and that also compresses QKV layers. Second, we look at a model with the same setup, except without compressing QKV layers, but still compressing ∼31% of the overall total FLOPs. Third, we look at a model that compresses both QKV and MLP layers but does not leverage the FLOP allocation procedure for MLP layers.

| Model Version | FLOP Compression Rate | PPL |
|---|:---:|:---:|
| Llama2-7b - MLP + QKV + FLOP Allocation | 31% | 8.40 |
| Llama2-7b - MLP + FLOP Allocation | 31% | 8.79 |
| Llama2-7b - MLP + QKV (No FLOP Allocation) | 32% | 9.10 |

Table 3: Perplexity Evaluation for Different RaNA Settings.

As we can observe in Tab. 3, the best model is the one combining both MLP + QKV compression and the FLOP allocation procedure in the MLPs, while the two other versions fall behind in perplexity.

**Accuracy-FLOPs trade-off including SliceGPT:**

In Fig. 5, we include the SliceGPT curve for the accuracy-FLOPs trade-off in Llama2-7b. There, we can observe that RaNA and CATS achieve a better accuracy-FLOPs trade-off than SliceGPT across various FLOP compression rates. Notably, RaNA and CATS are adaptive compression methods designed to reduce FLOPs rather than memory usage; hence, we focused our analysis on FLOP compression rates. On the other hand, SliceGPT is a static compression method that targets both FLOP and memory reductions, with their work principally discussing their slicing compression rate, which tracks memory savings. We believe the observed performance gap is attributed to this fundamental trade-off between static and adaptive compression methods: static compression methods seek to attain both memory and FLOPs savings, at the cost of a less favorable accuracy-FLOPs trade-off compared to adaptive approaches.

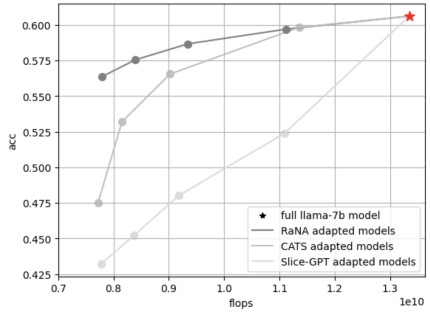

Figure 5: Llama2-7b Accuracy v.s. FLOPs.

## A.3 FLOP COMPRESSION BREAKDOWN

Here we leave in Tab. 4 the FLOP compression breakdown across MLP and QKV layers of the main adapted models.

| Model | Total FLOP Compression | MLP FLOP Compression | QKV FLOP Compression |
|---|---|---|---|
| Gemma-2b-RaNA | 44% | 61% | 0% |
| Gemma-2b-CATS | 47% | 65% | 0% |
| Gemma-2b-RaNA | 32% | 45% | 0% |
| Gemma-2b-CATS | 34% | 48% | 0% |
| Gemma-2b-RaNA | 19% | 27% | 0% |
| Gemma-2b-CATS | 20% | 28% | 0% |
| Llama2-7b-RaNA | 42% | 47% | 46% |
| Llama2-7b-CATS | 42% | 65% | 0% |
| Llama2-7b-RaNA | 30% | 34% | 33% |
| Llama2-7b-CATS | 32% | 50% | 0% |
| Llama2-7b-RaNA | 17% | 19% | 18% |
| Llama2-7b-CATS | 15% | 23% | 0% |

Table 4: FLOP Compression Comparison for Different Models. Total, MLP, and QKV FLOP compression rates are reported for the Gemma-2b and Llama2-7b models with RaNA and CATS adapters.

A.4  ALGORITHM

We provide a pseudocode implementation of RaNA below:

---

**Algorithm 1** RANA Layer Compression

---

1: **procedure** COMPRESSLAYER(layer, decomposition, prune_ratio)
2:     // Find optimal rank and thresholds for compression
3:     $rank, threshold \leftarrow$ ComputeOptimalParameters($decomposition, prune\_ratio$)
4:     // Create compressed layer using low-rank approximation
5:     $compressed\_layer \leftarrow$ DecomposeToRankN($layer, rank$)
6:     // Apply adaptive thresholding for dynamic sparsity
7:     $masked\_layer \leftarrow$ ApplyThresholdMask($compressed\_layer, threshold$)
8:     **return** $masked\_layer$
9: **end procedure**

---

**Algorithm 2** RANA MLP Transformation

---

1: **procedure** TRANSFORMMLP(mlp, input_data, prune_ratios)
2:     // Transform each component with different pruning ratios
3:     $up\_masked \leftarrow$ CompressLayer($mlp.up\_proj, prune\_ratios.up$)
4:     $gate\_masked \leftarrow$ CompressLayer($mlp.gate\_proj, prune\_ratios.gate$)
5:     // Compute activation threshold for downstream pruning
6:     $act\_threshold \leftarrow$ ComputeActivationThreshold($mlp, input\_data, prune\_ratios.down$)
7:     // Construct efficient MLP with dynamic sparsity
8:     $efficient\_mlp \leftarrow$ CreateDynamicMLP($up\_masked,$
9:         $gate\_masked, mlp.down\_proj, act\_threshold$)
10:    **return** $efficient\_mlp$
11: **end procedure**

---

**Algorithm 3** RANA Forward Pass

---

1: **procedure** FORWARDPASS(input, compressed_layer)
2:     // Dynamic pruning based on activation magnitudes
3:     $activations \leftarrow$ ComputeActivations($input$)
4:     $pruned\_activs \leftarrow$ ApplyDynamicMask($activations$)
5:     // Efficient forward computation using compressed weights
6:     $output \leftarrow$ ComputeOutput($pruned\_activs$)
7:     **return** $output$
8: **end procedure**

---

