# OpenReview forum: "Adaptive Rank Allocation: Speeding Up Modern Transformers with RaNA Adapters"
_ICLR.cc/2025/Conference — ICLR 2025 Poster_

### Official Review · Reviewer_5oY9 · 2024-11-02

**Soundness:** 3
**Presentation:** 3
**Contribution:** 3
**Rating:** 6
**Confidence:** 3

**Summary:**

This paper introduces a dynamic rank allocation method for improving the computational complexity of a transformer model.

The paper describes a related body of work on neural adaptors, and recognizes that those that rely on sparse activation functions (ReLU) are hardly suitable for transformers, the majority of which now use non-sparse activations functions (e.g. SwiGLU). Neural adaptors that do not rely on sparse activations are better suited to transformers but suffer from inefficiencies, primarily because of the need to densely evaluate the output of the gating layer. Besides, these methods leave the question of optimizing attention layers (QKV projection) unanswered.

The paper further sets context with adaptive rank allocation, which consists in decomposing a linear into two lower-rank layers parameterized by A and B matrices. Importantly, the rank is a function of the input x, and is determined by a learnable router r(x). The paper notes that adaptive rank allocation is a generalization of neural adaptors, which can be reformulated as adaptive rank allocation.
Linear layer rank adaptors can be simplified if the router is not input dependent, in which case the determination of an optimial decomposition into and B can be performed using the Eckart-Young theorem.
Further to this, a binary masker may be derived to output an input-dependent mask which, in expectation, leads to the target average rank. Specifically the binary masker requires calculating (Bx)^2. This operation is inexpensive when the output dimension is bigger than the input dimension and is thus suitable for QKV, up-projection and gate-projections.
The paper then tackles the case of down-projections: for these layers, a simple neural thresholding masker is used.

Experimental results are shown to empirically prove that a small number of A column vectors capture most of the information, making low-rank approximation efficient.
Further experimental results show superior accuracy and perplexity vs other methods at similar compression ratios.

**Strengths:**

The paper is very well written and is grounded with elaborate theory on the subject.
The topic of the paper is important for the field of transformers.
The method could be applied to a broad range of model architectures, including vision transformers.

**Weaknesses:**

The paper compares the method against a rather narrow range of baselines. It would be interesting to compare against pruning baselines, as these are related methods for optimizing the runtime performance of models.  Other baselines could include LLMPruner, SliceGPT, ShearedLLaMA, FlexTron, Minitron, etc.

The paper does not give much details on the practical implementation of the method. It would be useful to see more information on the training process. It would help readers to have an informative diagram to summarize the process. It would also be interesting to know how the method interfaces with DL frameworks (PyTorch, FlexAttention, etc.). An open-source implementation would be best.

**Questions:**

Can you provide a pseudo-code implementation of the method?

How does the method compare against FlexTron (https://arxiv.org/pdf/2406.10260), Minitron (https://arxiv.org/pdf/2408.11796).

What is the relationship between the neural thresholding employed for down-projection layers and CATS?

Are the compression ratios mentioned in experimental results taking the contribution of the router FLOPs into account? Can you show a breakdown of total FLOPs between the main computation and the computation of the routers.

How is the FLOPs reduction achieved in practice? Are tensor sliced and how does this affect memory caching/coalescing? Which DL framework was used to implement the method?

Does batched inference work and achieve the desired speed-ups if masks are inconsistent between samples?

Could you share end-to-end inference latency measurements and compare the achieved speed-up with the FLOPs reduction?

---

> ### Author Response · Authors · 2024-12-02
> **Response to Reviewer 5oY9 (Part 1)**
>
> We thank reviewer 5oY9 for their positive comments and helpful constructive feedback! We note that the wall-clock latency measurements and comparisons to other related work suggestions are addressed in the general response above. Further, we reply to specific suggestions below.
>
> **W1/Q2: Lack of baselines from the structured pruning literature and comparison to other compression methods.**
>
> Thanks for bringing up this point! In Section 5 of our updated manuscript, we include additional comparisons to SliceGPT [2], a cutting-edge open source pruning method, which was the most suggested by reviewers (Table 1, Figure 5). Please refer to the general response for more details.
>
> **W2/Q1. Adding practical implementation details**
>
> Thanks for requesting this! In our revised manuscript, we now incorporate implementation pseudocode in the appendix (sect. 7.2).
>
> **Q3. What is the relationship between the neural thresholding for down-projection layers and CATS?**
>
> This is a good question, thanks for bringing it up! The neuron masker method is indeed slightly different from the one CATS used. To be concise:
> * The CATS neuron mask is $m_{CATS}(x) _i= |SiLU(GateProj(x))_i| >= t$.
> * The RaNA neuron mask is $m_{RaNA}(x)_i = |SiLU(GateProj(x))_i \cdot UpProj(x)_i| \cdot ||W^{down}_i||_F >= t$. (Eqn. 12)
>
> We argue this masker is more natural. Notably our proposed masker looks at $SiLU(GateProj(x))_i \cdot UpProj(x)_i$ since it is the input to the down-project matrix. Masking out based only on $SiLU(GateProj(x)) _i$ may be troublesome if $SiLU(GateProj(x))_i \cdot UpProj(x)_i$ results in a large value for the masked out entry. Moreover incorporating the $||W^{down}_i||_F$ is more natural if we are considering reducing the overall output error of the MLP.
>
> **Q4. Are compression ratios including router FLOPs? Can you show a breakdown of FLOPs distribution between main computation and maskers?**
>
> Thanks for this detailed question! Yes, the masker FLOPs are considered into the total FLOPs. Nevertheless, we note that element-wise operations over vectors were generally not considered in the FLOP computation.
>
> We’d like to note that it's unclear how to devise a breakdown between main computation and masker FLOPs for CATS and RaNA, since the masker can be thought of as adding an elementwise operation on top of an intermediate variable that will be used in computing the output anyways. For example, for the linear-layer rank adapter masker (eqn. 9), for the masker we look at the output of $Bx$, element-wise square it and compare each element to a threshold. However, $Bx$ is still used in the overall computation anyway (eqn. 4). For this reason, in the case of RaNA and CATS, it is unclear how to determine what FLOPs belong to the main computation and which belong to the masker, since they are heavily dependent.
>
> For the Pythia suite models, the breakdown is possible for our neuron-adaptive baseline. For the neuron-adaptive baseline, the masker is described as the MLP sigmoid masker from Sect. 4.1. As outlined in Section 5.1, we allocated 6% of the MLP FLOPs to the masker and the rest to the main computation, as done in [3].
>
> **Q5.1. How is the FLOP reduction achieved in practice?**
>
> Thanks for this question! FLOP reduction is achieved by applying a sparse mask to each linear layer in the MLP and QKV projection layers. For example, from Eq. 11, the Up-Project, Gate-Project and QKV layers are approximated using a linear-layer rank adapter, $\text{Linear}(x) \approx  \text{Linear'}(x) = A (m(x) \odot B x )$, where $m(x) \in \\{0,1\\}^r$ is a sparse binary mask. After computing $v=m(x) \odot B(x)$, $v$ becomes sparse, allowing us to skip the product involving the columns of $A$ corresponding to the 0 entries in $m(x)$. In a similar fashion, the neuron-adapter leveraged in the Down-Project layers constitutes a matrix-vector product, where the vector is sparse, allowing us to skip computations. Moreover, we realize practical speedups associated with this theoretical FLOP reduction by leveraging a masked GEMV kernel, as we detail in the general response.
>
> **Q5.2. Which DL framework was used to implement the method?**
>
> Thanks for bringing this up! The method is implemented in PyTorch, as clarified in section 5.1 from the revised manuscript. Moreover, the efficient version of RaNA leverages a masked GEMV kernel.

---

> ### Author Response · Authors · 2024-12-02
> **Response to Reviewer 5oY9 (Part 2)**
>
> **Q6. Does batched inference work and achieve speed-ups if masks are inconsistent?**
>
> This is a good question, thanks for asking it! Our method and experiments focused on single-sequence decoding (batch size = 1). We found the batch size = 1 decoding scenario relevant due to its importance in systems where latency matters more than throughput (e.g. on-device workloads). Notably, due to their input dependent nature, neuron-adaptive adapters like CATS also mainly look at this setting. Therefore, we decided to leave the larger batch size scenario outside the current scope of our work. However, leveraging a masked GEMV for larger batch sizes and measuring latency would be an interesting direction of future work.
>
> **Q7: Lack of wall-clock latency measurements for RaNA.**
>
> Thank you for raising this point! In the revision, we leveraged an efficient implementation of RaNA using a masked GEMV Triton kernel and included a wall-clock comparison of RaNA and CATS [1] (Figure 1b). Please refer to the general response for more details.
>
> **References:**
> 1. Lee, Je-Yong, et al. “Contextually-aware thresholding for sparsity in large language models”, 2024. URL https://arxiv.org/abs/2404.08763.
> 2. Ashkboos, Saleh et al. “Slicegpt: Compress large language models by deleting rows and columns”, 2024. URL https://arxiv.org/abs/2401.15024.
> 3. Zhang, Zhengyan et al. “Relu2 wins: Discovering efficient activation functions for sparse llms, 2024”. URL https://arxiv.org/abs/2402.03804.

---

### Official Review · Reviewer_yoCw · 2024-11-03

**Soundness:** 2
**Presentation:** 3
**Contribution:** 3
**Rating:** 6
**Confidence:** 3

**Summary:**

The work proposes a new method for input-dependent compute allocation in transformer layers that is not dependent on activation sparsity and can be applied to any linear layer. The idea is to decompose $Wx$ as $A(r(x) \odot Bx)$ and find $A$, $B$ and $r$ such that the approximation error of $Wx$ is minimal. The derived 'RaNA'-adapters are  tested on MLP and QKV layers in a variety of Transformer models, demonstrating reduced approximation errors as well as improved performance on downstream tasks when compared to conventional neuron adapter baselines such as CATS (Lee+2024).

**Strengths:**

The key idea of this work to apply adaptive compute at the rank level of linear layers makes for a straightforward and generally applicable sparsification strategy that unlike other prior methods does not require specific architectural features or activation functions.

The formulation as an error minimization problem over the layer input distribution allows for a principled derivation of suitable decomposition and routing functions.

The experimental results show improvements over existing neuron-adaptive methods.

**Weaknesses:**

While the experiments show that RaNA outperforms CATS in perplexity and accuracy, the setup offers little insight into where this improvement comes from. Table 1 compares RaNA with sparse MLP/QKV against CATS with sparse MLP only. Given that they are compared at an equal compression rate, I assume that the MLPs in the CATS model are more aggressively sparsified than the MLPs in RaNA. This makes it hard to conclude what causes the performance difference. Table 2 directly compares MLP-only sparsification but is for a different, smaller model Gemma-2b. It would be helpful to add RaNA with MLP-only sparsification for Llama2-7b to differentiate the influence of MLP vs. QKV sparsification. Furthermore, it would be useful to add an additional baseline like LLRA that like RaNA can sparsify MLP+QKV. To aid the comparison, could you also clarify the sparsification levels for each component (MLP, QKV) in both RaNA and baseline methods?

L509 offers different hypotheses for the causes of RaNA's performance (compression capacity, FLOP distribution, QKV applicability) but an ablation study to investigate the factors is missing. Adding results that can shed light on RaNA's performance could greatly strengthen the analysis, for instance, by comparing RaNA with and without QKV sparsification, or with different FLOP distribution strategies.

I suggest moving the proof for Proposition 1 into the main text and describing it as a concrete example to help the reader understand how the neuron-adaptive method is a special case of rank adaption. Specifically, I found it helpful to see that $B_{down}, A_{up}, r_{up} := I$ and $A_{down}, B_{up}$ adjustment are all it takes to get to the neuron-adapted version of the MLP.

**Questions:**

When describing the downstream and perplexity results in L480, you state that QKV is only RaNA-fied for Llama and Pythia, but not Gemma. Could you please explain why RaNA was not applied to QKV in Gemma?

---

> ### Author Response · Authors · 2024-12-02
> **Response to Reviewer yoCw**
>
> We thank reviewer yoCw for their positive comments and helpful constructive feedback! We note that the suggestion on comparisons to other related work (specifically a method that also compresses MLP + QKV layers for Llama2) is addressed in the general response above. Further, we reply to specific suggestions below.
>
> **W1. Insights on MLP compression v.s. MLP + QKV compression and using different FLOP allocation strategies in RaNA.**
>
> This is a great question, thank you for raising it! In Section A.3, we provide an ablation study where we compare the perplexity of RaNA, before fine-tuning, on 3 different settings to assess the performance of the adapter when compressing different layers or using different FLOP allocation strategies. Concretely, we observed the RedPajama perplexity on a RaNA-adapted Llama2-7b model under the following settings, all targeting the same total FLOP compression ratio of ~31%:
> * Vanilla RaNA adapter compressing MLPs + QKVs, leveraging the FLOP allocation strategy
> * RaNA adapter only compressing MLPs, leveraging the FLOP allocation strategy
> * RaNA adapter compressing MLPs + QKVs, uniformly allocating FLOPs across all the MLP components (Up-Project, Gate-Project and Down-Project layers).
>
> Notably, the best adapted model was the one leveraging the vanilla RaNA adapter (8.40 ppl), with the other two settings falling behind in perplexity (8.79 ppl and 9.10 ppl). Moreover, the FLOP allocation strategy showed to be of high importance as the adapted model took a notable performance hit when ablating this component (0.7 ppl points).
>
> **W2. Comparison against baseline that sparsifies MLP and QKV layers like RaNA**
>
> Thanks for this suggestion! In Section 5 of our updated manuscript, we include additional comparisons to SliceGPT [2], a cutting-edge pruning method (Table 1, Figure 5). Please refer to the general response for more details.
>
> **W3. Clarify sparsification levels for each component (MLP, QKV) in RaNA and baseline methods.**
>
> This is a good point, thanks for bringing it up! Definitely, we added a breakdown of sparsification levels for MLP and QKV layers in RaNA and baseline methods in Table 4.
>
> **W4. Move the proof for Proposition 1 into the main text**
>
> Thanks for pointing this out! We are glad you found the proof helpful. We have moved the proof to Section 3 in the revised manuscript.
>
> **Q1. Why was RaNA not applied to QKV in Gemma?**
>
> Thanks for this question, this is a great observation! RaNA was not applied to QKV in Gemma since we did not expect much speed up from compressing those layers, as they consume much less flops than the MLPs. Concretely, the Query, Key and Value projection together consume only ~5% of the FLOPs relative to those consumed by MLPs in Gemma-2b, while they consume ~37% in the case of Llama2-7b. Therefore, we focused on MLP sparsification for Gemma. In our revision, we have added this clarification in Section 5.3 of the paper.
>
> **References:**
> 1. Lee, Je-Yong, et al. “Contextually-aware thresholding for sparsity in large language models”, 2024. URL https://arxiv.org/abs/2404.08763.
> 2. Ashkboos, Saleh et al. “Slicegpt: Compress large language models by deleting rows and columns”, 2024. URL https://arxiv.org/abs/2401.15024.

---

> > ### Comment · Reviewer_yoCw · 2024-12-02
> >
> > Thanks for your detailed response. I appreciate the author's efforts to incorporate additional results and clarifications into the manuscript. Overall, the new baselines and benchmarks strengthen the case that RaNA offers a viable trade-off for speeding up transformer computations. In the light of this, I am willing to increase my score from 5 to 6.

---

### Official Review · Reviewer_LZp1 · 2024-11-04

**Soundness:** 2
**Presentation:** 3
**Contribution:** 2
**Rating:** 6
**Confidence:** 3

**Summary:**

This paper presents a novel model compression framework that leverages adaptive rank allocation via rank and neuron allocation (RaNA) adapters. Specifically, weights in linear layers are decomposed into two low-rank matrices A and B, with a learnable router in between that selects which ranks to activate. RaNA also supports adaptive FLOP allocation across adapters. The paper evaluates RaNA on LLMs such as Llama2-7B and Gemma-2b, and compares its performance to other recent neuron adaptation methods such as CATS. The evaluation compares local (layer-wise) reconstruction errors for RaNa vs. baselines, FLOPs, and accuracy on downstream tasks.

**Strengths:**

* The paper is reasonably well-written and easy to follow.
* Model compression is a promising way of reducing LLM size and making it fit various deployment constraints such as target latency, parameter and memory-footprint. As such, this paper targets a relevant and important problem.

**Weaknesses:**

* I’m not a big fan of using FLOPs as an optimization metric - it is well-known at this point that FLOP reductions may or may not correspond to reductions in inference latency or wall-clock time. This is likely also the reason why the main baseline method used in the paper (CATS) introduces a custom GPU kernel to realize runtime speedups. Have the authors done any study on how the proposed method improves the inference latency or throughput of the decomposed network? A theoretical discussion may also suffice given the limited time.
* I believe that RaNA needs to be compared to other related work on structured pruning to better understand its performance and specific scenarios where it does well. Examples might include Sheared Llama [1], ShortGPT [2] , SliceGPT [3], or Minitron [4].
* The size (< about 7B parameters) and age (Llama2 was released nearly 2 years ago at this point) of the models being used for evaluation is a weakness of the paper. While I understand that scaling up to larger models is not always feasible given limited resources, adding results on newer models (eg: Llama3.1 8b) would help compare with other recent pruning/compression efforts.

References:

1. Xia, Mengzhou, et al. "Sheared llama: Accelerating language model pre-training via structured pruning." arXiv preprint arXiv:2310.06694 (2023).
2. Men, Xin, et al. "Shortgpt: Layers in large language models are more redundant than you expect." arXiv preprint arXiv:2403.03853 (2024).
3. Ashkboos, Saleh, et al. "Slicegpt: Compress large language models by deleting rows and columns." arXiv preprint arXiv:2401.15024 (2024).
4. Muralidharan, Saurav, et al. "Compact language models via pruning and knowledge distillation." arXiv preprint arXiv:2407.14679 (2024).

**Questions:**

* Please address the wall clock speedup comment in the weaknesses section.
* On the topic of inference speedups, RaNA replaces a single GEMM in linear layers with two GEMMs (please correct me if I’m wrong) - I’d really like to understand how this might affect real-world inference latency.
* How well does RaNA perform on networks with sparse activation functions?

---

> ### Author Response · Authors · 2024-12-02
> **Response to Reviewer LZp1**
>
> We thank reviewer LZp1 for their positive comments and helpful constructive feedback! We note that the wall-clock latency measurements and comparisons to other related work suggestions are addressed in the general response above. Further, we reply to specific suggestions below.
>
> **Q1/W1: Lack of wall-clock latency measurements for RaNA.**
>
> Thank you for raising this point! In the revision, we leveraged an efficient implementation of RaNA using a masked GEMV Triton kernel and included a wall-clock comparison of RaNA and CATS [1] (Figure 1b). Please refer to the general response for more details.
>
> **W2: Lack of baselines from the structured pruning literature.**
>
> Thank you for this important point! In Section 5 of our updated manuscript, we include additional comparisons to SliceGPT [2], a cutting-edge pruning method (Table 1, Figure 5). Please refer to the general response for more details.
>
> **W3. Results on newer models (eg: Llama3.1 8b) would help compare with other recent pruning/compression efforts.**
>
> Thank you for bringing this point up! Unfortunately, we were limited in resources, which did not permit us to test RaNA in bigger models. Nevertheless, CATS and SliceGPT were both applied to Llama2-7b models, which motivated us to use it as our largest model for testing. Therefore, given the limited timing, we decided to keep our current analysis for Llama2 models only. However, we hope to look into assessing the performance of RaNA in Llama3 for the camera ready version.
>
> **Q2. How does replacing a single GEMM with two GEMMs in linear layers affect latency?**
>
> Thank you for this question! That’s correct. RaNA replaces a single general matrix vector multiply (GEMV) with two for Up-Project, Gate-Project and QKV linear layers. Concretely, the linear layers adapted using the linear-layer rank adapter of RaNA replace a single GEMV with one vanilla GEMV for $v=Bx$ and one masked GEMV for $A(m(x) \odot v)$.
>
> Our experiments show that RaNA achieves practical inference speedups across various compression ratios and hint that decomposing a single GEMV with two does not impact performance drastically. Concretely, from Figure 1.b we can see that, as latency increases, RaNA’s downstream task performance converges to that of the full model, with only a ~0.85 percentage-point gap at ~40ms/token compared to the full model's ~46ms/token latency. These results show that despite decomposing a matrix multiplication into two operations, practical speedups are attained. Furthermore, the convergence of the downstream-task performance towards the full model performance as latency increases hints that the decomposition does not drastically impact speed.
>
> **Q3. How well does RaNA perform on networks with sparse activation functions?**
>
> Thanks for bringing this up! We would like to emphasize that the main objective of the RaNA adapters, as outlined in our introduction, is to improve adaptive compression for modern transformer architectures leveraging non-sparse activations. Notably, previous work has found conventional neuron adapters to work well for sparse activation functions, however, a lack of performance has been noticed when applied to non-sparse activation functions [3]. This has led to research on tackling this specific problem, for example ReLUfication [4], and CATS [1]. Our goal with RaNA aligns with previous work, as we seek to improve adaptive compression for modern Transformers with non-sparse activation functions. Therefore, for this work, we decided to not assess our method in sparse activation functions, as we determined it out of scope for the problem we are trying to address.
>
> **References:**
> 1. Lee, Je-Yong, et al. “Contextually-aware thresholding for sparsity in large language models”, 2024. URL https://arxiv.org/abs/2404.08763.
> 2. Ashkboos, Saleh et al. “Slicegpt: Compress large language models by deleting rows and columns”, 2024. URL https://arxiv.org/abs/2401.15024.
> 3. Zhang, Zhengyan et al. “Relu2 wins: Discovering efficient activation functions for sparse llms, 2024”. URL https://arxiv.org/abs/2402.03804.
> 4. Mirzadeh, Iman et al. “Relu strikes back: Exploiting activation sparsity in large language models”, 2023. URL https://arxiv.org/abs/2310.04564.

---

> > ### Comment · Reviewer_LZp1 · 2024-12-02
> >
> > I'd like to thank the authors for addressing most of the questions and concerns in my original review. I further commend them for adding new results on runtime performance, and comparing RaNA to SliceGPT in the limited rebuttal time-frame. I'll raise my score to 6.

---

### Official Review · Reviewer_RtUt · 2024-11-04

**Soundness:** 2
**Presentation:** 3
**Contribution:** 2
**Rating:** 6
**Confidence:** 4

**Summary:**

Authors propose RaNA (Rank and Neuron Allocator), aiming to timprove the efficiency of LLMs by compressing linear layers. Authors propose the adaptive rank allocation, and decompose all linear layer into a product of low-rank matrices and an adaptive router.

**Strengths:**

1. The paper is clearly written.
2. The method is applicable to all linear layers.
3. Error achieved by RaNA is far smaller than previous methods and SVD.

**Weaknesses:**

1. no wall-clock latency measurement. Measuring only FLOPs is not acceptable for a compression method on transformers.
2. The paper lacks baselines from the pruning literature (WANDA, SliceGPT). These are baselines that must be compared for this method since RaNA achieved compression by low rank and sparsity.
3. The proposed method takes individual layers into account while deciding sparsity. Some literature in LoRA (AdaLoRA) shows that different layers may need different ranks in the adaptation context. What is the gap between considering only one layer’s error and considering multiple layers’ errors while determining the rank to mask out?
============ post rebuttal ==============
Authors resolved my concerns. I will raise my score to 6.

**Questions:**

See weaknesses

---

> ### Author Response · Authors · 2024-12-02
> **Response to Reviewer RtUt**
>
> We thank reviewer RtUt for their positive comments and helpful constructive feedback! We note that the wall-clock latency measurements and baseline suggestions are addressed in the general response above. Further, we reply to specific suggestions below.
>
> **W1: Lack of wall-clock latency measurements for RaNA.**
>
> Thank you for raising this point! In the revision, we leveraged an efficient implementation of RaNA using a masked GEMV Triton kernel and included a wall-clock comparison of RaNA and CATS [1] (Figure 1b). Please refer to the general response for more details.
>
> **W2: Lack of baselines from the structured pruning literature.**
>
> Thank you for this important point! In Section 5 of our updated manuscript, we include additional comparisons to SliceGPT [2], a cutting-edge pruning method (Table 1, Figure 5). Please refer to the general response for more details.
>
> **W3: What is the gap between considering only one layer’s error and considering multiple layers’ errors while determining the rank to mask out?**
>
> This is a great question, thank you for bringing it up! We acknowledge that considering multiple layers’ errors could potentially yield better results, as prior work suggests that some layers contribute more significantly to overall performance [3]. In this work, we are focused on the problem of extending neuron-adaptive methods to non-sparse activations. As such, we chose to focus on the simpler setting of layer-wise optimization rather than holistic optimization across layers, which introduces additional complexity. We recognize that a holistic FLOP allocation is an interesting direction to explore on improving RaNA, which we hope to investigate in future work.
>
> **References:**
> 1. Lee, Je-Yong, et al. “Contextually-aware thresholding for sparsity in large language models”, 2024. URL https://arxiv.org/abs/2404.08763.
> 2. Ashkboos, Saleh et al. “Slicegpt: Compress large language models by deleting rows and columns”, 2024. URL https://arxiv.org/abs/2401.15024
> 3. LLMP: Ma, Xinyin, et al. “Llm-pruner: On the structural pruning of large language models”, 2023. URL https://arxiv.org/pdf/2305.11627.

---

> > ### Comment · Reviewer_RtUt · 2024-12-02
> > **Inconsistent SliceGPT results with original SliceGPT paper**
> >
> > My thanks to the authors for all the additional results.
> >
> > I checked SliceGPT's(table 8) result [1] on ARC-e and ARC-c. LLaMA2-7b's result on ARC-e and ARC-c is 63.93% and 37.80%, respectively, at 30% sparsity. The authors' updated result is 48.19% and 26.19%, respectively, at 31% sparsity.
> >
> > I encourage authors to either debug their SliceGPT implementation or explain this ~20% discrepancy with the original paper.
> > [1] Ashkboos, Saleh et al. “Slicegpt: Compress large language models by deleting rows and columns”, 2024. URL https://arxiv.org/abs/2401.15024.

---

> > > ### Author Response · Authors · 2024-12-03
> > > **Response to SliceGPT Observation**
> > >
> > > Thank you for this observation! This is an important point which is worth further discussing in the manuscript. We would like to clarify the following important points regarding this comparison:
> > >
> > > 1. **Different compression metrics: slicing vs. FLOPs.** While the SliceGPT paper \[1] reports compression via the percentage of parameters sliced off, we report the reduction in FLOPs.
> > >
> > > 2. **Different datasets: Alpaca vs. RedPajama.** Table 8 from \[1] performs slicing using the Alpaca (instruction-tuning) dataset, whereas, mirroring the compression procedures we use for RaNA and CATS, we use the RedPajama (corpus) dataset. Section 4.1 of \[1] notes: "We see a marked difference between RFT on WikiText-2 and Alpaca datasets, with the Alpaca dataset giving much higher performing models. We attribute this difference to the similarity between Alpaca and the benchmark tasks.". We believe this difference between instruction-tuning vs. pre-training corpus datasets contributes significantly to the gap in downstream accuracy. Furthermore, Table 8 reports slicing only (no finetuning), whereas we perform both slicing and finetuning. As such, we believe a more appropriate comparison would be with Table 9 of \[1], which reports slicing and finetuning on *WikiText-2*, another corpus dataset.
> > >
> > > In the table below, we provide a comparison of the reported SliceGPT performance from Table 9 of \[1], sliced and fine-tuned using WikiText-2, vs. our SliceGPT implementation, sliced and fine-tuned using RedPajama. We report both slicing and FLOPs-based compression metrics. We observe that our results closely match the reported performances: our implementation performs slightly better on ARC-e and slightly worse on ARC-c, but performance gaps drop to around 6% or less. Note that this level of variation is smaller than the reported gap between slicing and fine-tuning using WikiText-2 vs. Alpaca (Table 10), which we also report below.
> > >
> > > |                                           |                   |                   |           |           |
> > > | ----------------------------------------- | ----------------- | ----------------- | --------- | --------- |
> > > |                                           | Slice compression | FLOPs compression | ARC-e     | ARC-c     |
> > > | SliceGPT, WikiText-2 (Table 9 of \[1])    | 30%               | 21%               | 50.76     | 34.13     |
> > > | SliceGPT, WikiText-2 (Table 9 of \[1])    | 25%               | 15%               | 52.36     | 35.75     |
> > > | **SliceGPT, RedPajama (Table 1 of ours)** | **27%**           | **17%**           | **58.29** | **31.31** |
> > > | SliceGPT, Alpaca (Table 10 of \[1])       | 30%               | 21%               | 66.54     | 40.87     |
> > > | SliceGPT, Alpaca (Table 10 of \[1])       | 25%               | 15%               | 68.22     | 42.83     |
> > >
> > > We additionally note that remaining differences in performance might be attributed to other details in fine-tuning procedures like learning rate selection (discussed in general response), length of finetuning, or generally other hyperparameter choices. We commit to provide more details about the comparison of SliceGPT in the camera-ready version.
> > >
> > > 1. Ashkboos, Saleh et al. “Slicegpt: Compress large language models by deleting rows and columns”, 2024. URL[ https://arxiv.org/abs/2401.15024](https://arxiv.org/abs/2401.15024)

---

### Author Response · Authors · 2024-12-02
**General Response**

We thank the reviewers for their constructive feedback and their thoughtful evaluation of our submission. We are encouraged to see that reviewers found the problem relevant and impactful (LZp1, 5oY9), our method to be general and widely applicable (RtUt, yoCw, 5oY9), and our writing to be clear (RtUt, LZp1, 5oY9). Additionally, we appreciate the recognition of the theoretical grounding (5oY9, yoCw) of RaNA.
To address the reviewers' suggestions, we have revised the manuscript and conducted additional experiments: 1) We measured wall-clock latency and demonstrated that RaNA presents improvements on the downstream-task performance to latency tradeoff (up to ~7 percentage-points in the 31-32 ms/token latency regime) compared to CATS [1]. 2) We added a cutting-edge baseline from the pruning literature, namely SliceGPT [2], to our analysis and showed that RaNA outperforms it in downstream task performance even in the small compression regimes, where it realizes an ~8 percentage-point improvement. 3) We performed an ablation study, where we looked at the performance boosts from applying RaNA to MLP + QKV layers vs. to MLP layers only vs. to QKV + MLP layers without the FLOP allocation strategy.

These updates are reflected in the revised manuscript. We also address the suggestions brought up by each reviewer in the individual responses below.

**Wall-Clock Latency and Practical Speedups**

*Suggested by: RtUt, LZp1, 5oY9*

Multiple reviewers suggested that we measure the wall clock latency of RaNA to assess its practical speedups. In the revised manuscript, we measure and compare the wall-clock latency of RaNA and CATS [1] at various FLOP compression rates (the results are shown in Fig. 1b). Our results demonstrate that RaNA’s theoretical FLOPs reduction also corresponds to real-world performance gains. Notably, RaNA presents a consistently better downstream-task performance to latency trade-off, with its tradeoff curve lying above CATS’ throughout. Moreover, at lower latency settings, RaNA attains considerably better accuracies: for example, at the 31-32 ms/token latency, RaNA achieves an average accuracy of ~54%, while CATS attains a ~47% average accuracy.

To realize these speedups, we implemented RaNA in PyTorch and further leveraged a masked GEMV kernel implemented in Triton, which we obtained from CATS. Such a kernel efficiently skips masked-out computations in our adapters. Concretely, we use the masked GEMV in the linear-layer rank adapter and neuron-adapter components of RaNA, as both of these operations involve a masked matrix-vector multiply. In the case of the linear-layer rank adapters, we leverage the masked GEMV in computing the matrix-vector product $A (m(x) \odot v)$, and in the neuron-adapter in the matrix-vector product $W (m(x) \odot x)$.


**Additional Baselines**

*Suggested by: All Reviewers*

All reviewers suggested that we compare the dynamic pruning methods studied in our work (RaNA and CATs) to static structured pruning methods. In the revision, we include comparisons with an open source cutting-edge static pruning method (and the most suggested method by reviewers), namely SliceGPT, for Llama2-7b. Results show that both RaNA and CATS consistently outperform SliceGPT, even in the smallest compression regimes (Table 1, Figure 5). For example, for a 17% compression, RaNA outperforms SliceGPT by margins of ~8 percentage-points in accuracy and ~4 perplexity points. This analysis highlights the robustness of adaptive methods like RaNA and CATS compared to static structured pruning approaches. Concretely, we note that CATS and RaNA are more powerful methods using an adaptive compute setup, which can be thought of determining input dependent weight matrices on the fly, as opposed to SliceGPT, which has static compressed weight matrices.

To elaborate, for this study, we sliced various SliceGPT [2] instances using the RedPajama dataset such that the resulting compressed models matched the FLOPs consumed by the RaNA adapted models when decoding a 512 token-long sequence. Further, we fine-tuned the sliced models on RedPajama for the same number of tokens as RaNA and CATS. Notably, SliceGPT, like RaNA, compresses both MLP and QKV layers. It’s worth mentioning that, due to time and compute constraints, we were only able to finetune SliceGPT instances in float16 precision and using a learning rate of 1e-5 (which we found to work well in preliminary experiments with float16 precision training), as outlined in the current manuscript. We commit to running more comprehensive comparisons for the camera-ready version of our work, including fine tuning SliceGPT in float32 precision with a proper sweep of LR.

**References:**
1. Lee, Je-Yong, et al. “Contextually-aware thresholding for sparsity in large language models”, 2024. URL https://arxiv.org/abs/2404.08763
2. Ashkboos, Saleh et al. “Slicegpt: Compress large language models by deleting rows and columns”, 2024. URL https://arxiv.org/abs/2401.15024

---

### Meta-Review · Area_Chair_GU6K · 2024-12-21

**Metareview:**

**Summary:** The paper introduces RaNA adapters to improve computational efficiency in LLMs by leveraging low-rank matrix decompositions and adaptive masking applied to linear layers. This approach overcomes limitations in existing neuron-adaptive methods, such as dependence on sparse activations. Empirical results show that RaNA achieves up to a 44% reduction in FLOPs while improving perplexity by up to 7 points.

**Strength:**

1. The proposed RaNA adapters are broadly applicable to both MLP and QKV layers, addressing a key limitation in existing neuron-adaptive methods.

2. The paper provides a theoretically grounded framework with extensive empirical evaluations, showcasing gains in both efficiency and accuracy.

**Weakness:**

1. Insufficient evaluation: The evaluation is limited to relatively small and older models (e.g., LLaMA2-7B), which restricts the generalizability of the results. Additionally, the manuscript does not provide wall-clock latency or real-device speed-up measurements, which are critical for assessing practical performance.

2. Missing baselines: The manuscript omits several important baselines, such as SliceGPT and Wanda, which are necessary for a comprehensive comparison.


**Reasons for the decision:**

Since this paper demonstrates strong theoretical analysis and empirical performance, and the authors have addressed concerns by including new baselines, wall-clock latency results, and more detailed ablation studies in the rebuttal, all reviewers are now in favor of accepting the paper. I concur with the reviewers' assessment.

**Additional Comments On Reviewer Discussion:**

During the rebuttal period, the common concerns raised by the reviewers and the corresponding author responses are provided as follows:

**1. Wall-Clock Latency and Practical Speedups (RtUt, LZp1, 5oY9):**

Reviewer Concerns: Reviewers highlighted the absence of wall-clock latency measurements to substantiate theoretical FLOP reductions.

Author Response: The authors provided wall-clock latency comparisons between RaNA and the baseline CATS. Results showed consistent speedups with competitive downstream task performance.

**2. Missing baselines (RtUt, 5oY9, LZp1):**

Reviewer Concerns: Reviewers requested comparisons with more baselines from the pruning literature, such as SliceGPT, Wanda, and ShearedLLaMA, to evaluate RaNA’s effectiveness across a broader spectrum of methods.

Author Response: The authors added comparisons with SliceGPT, demonstrating RaNA’s superiority in both perplexity and accuracy at similar compression rates.

**3. Limited model scale (LZp1):**

Reviewer Concerns: The evaluation was limited to smaller models like LLaMA2-7B, with no experiments on more recent or larger-scale architectures like LLaMA3-70B.

Author Response: The authors acknowledged resource constraints and committed to exploring larger models for future work.

**4. Clarity and writing improvements (yoCw, 5oY9):**

Reviewer Concerns: Some sections, such as the proof for Proposition 1 and the description of practical implementation, were unclear.

Author Response: The authors improved clarity by moving the proof into the main text and adding an implementation pseudocode.

Since the authors have addressed concerns by including new baselines, wall-clock latency results, and more detailed ablation studies in the rebuttal, its merits outweigh the remaining limitations. As such, I am inclined to accept this paper.

---

### Decision · Program_Chairs · 2025-01-22

Accept (Poster)